# Nanoapatites Doped and Co-Doped with Noble Metal Ions as Modern Antibiofilm Materials for Biomedical Applications against Drug-Resistant Clinical Strains of *Enterococcus faecalis* VRE and *Staphylococcus aureus* MRSA

**DOI:** 10.3390/ijms23031533

**Published:** 2022-01-28

**Authors:** Emil Paluch, Paulina Sobierajska, Piotr Okińczyc, Jarosław Widelski, Anna Duda-Madej, Barbara Krzyżanowska, Paweł Krzyżek, Rafał Ogórek, Jakub Szperlik, Jacek Chmielowiec, Grażyna Gościniak, Rafal J. Wiglusz

**Affiliations:** 1Department of Microbiology, Faculty of Medicine, Wroclaw Medical University, 50-376 Wroclaw, Poland; anna.duda-madej@umw.edu.pl (A.D.-M.); barbara.krzyzanowska@umw.edu.pl (B.K.); pawel.krzyzek@umw.edu.pl (P.K.); grazyna.gosciniak@umw.edu.pl (G.G.); 2Institute of Low Temperature and Structure Research, Polish Academy of Sciences, Okolna 2, 50-422 Wroclaw, Poland; p.sobierajska@intibs.pl (P.S.); j.chmielowiec@intibs.pl (J.C.); 3Department of Pharmacognosy and Herbal Medicines, Wroclaw Medical University, 50-556 Wroclaw, Poland; piotr.okinczyc@umw.edu.pl; 4Department of Pharmacognosy with the Medicinal Plant Garden, Medical University of Lublin, 20-093 Lublin, Poland; jaroslaw.widelski@umlub.pl; 5Department of Mycology and Genetics, University of Wroclaw, Przybyszewskiego 63, 51-148 Wroclaw, Poland; rafal.ogorek@uwr.edu.pl; 6Faculty of Biological Sciences, Botanical Garden, University of Wroclaw, Sienkiewicza 23, 50-525 Wroclaw, Poland; jakubsz@hotmail.com

**Keywords:** nanoapatite, noble metal ions, Ag^0^/Au^0^/Pd^0^ nanoparticles, biofilm, bacteria, MRSA, VRE, antimicrobial, cytotoxicity

## Abstract

The main aim of our research was to investigate antiadhesive and antibiofilm properties of nanocrystalline apatites doped and co-doped with noble metal ions (Ag^+^, Au^+^, and Pd^2+^) against selected drug-resistant strains of *Enterococcus faecalis* and *Staphylococcus aureus*. The materials with the structure of apatite (hydroxyapatite, nHAp; hydroxy-chlor-apatites, OH-Cl-Ap) containing 1 mol% and 2 mol% of dopants and co-dopants were successfully obtained by the wet chemistry method. The majority of them contained an additional phase of metallic nanoparticles, in particular, AuNPs and PdNPs, which was confirmed by the XRPD, FTIR, UV–Vis, and SEM–EDS techniques. Extensive microbiological tests of the nanoapatites were carried out determining their MIC, MBC value, and FICI. The antiadhesive and antibiofilm properties of the tested nanoapatites were determined in detail with the use of fluorescence microscopy and computer image analysis. The results showed that almost all tested nanoapatites strongly inhibit adhesion and biofilm production of the tested bacterial strains. Biomaterials have not shown any significant cytotoxic effect on fibroblasts and even increased their survival when co-incubated with bacterial biofilms. Performed analyses confirmed that the nanoapatites doped and co-doped with noble metal ions are safe and excellent antiadhesive and antibiofilm biomaterials with potential use in the future in medical sectors.

## 1. Introduction

Uncontrolled growth of bacteria on implantable materials, including hydroxyapatite (Ca_10_(PO_4_)_6_(OH)_2_, hereafter HAp), can lead to the formation of bacterial biofilms that make antibiotic treatment difficult or ineffective [1]. Bacterial adhesion to the surface is one of the first and most important stages of biofilm formation. Thus, limiting this stage is crucial in reducing the development of biofilm [2,3]. There are several possibilities to decrease adhesion, however, the latest reports highlight quorum quenching particles and modern anti-adhesive surfaces (containing noble metal nanoparticles or enzyme-inorganic hybrid nanoflowers) as the most promising ones [4,5]. Bacteria growing as biofilm produce multicellular aggregates embedded within an extracellular polymeric matrix (EPM). EPM is most commonly composed of polysaccharides, lipids, proteins, and extracellular DNA (eDNA) [6]. While water channels within biofilms may partially allow penetration of this structure by some antibiotics, e.g., fluoroquinolones, their presence does not always guarantee complete eradication of biofilms [7]. The spatial architecture of biofilm is functionally diverse. In deeper parts of the biofilm, there is a subpopulation of metabolically inactive cells known as persister cells. These cells are able to survive extremely high concentrations of antimicrobials and after the treatment turn into a metabolically active state and recreate the biofilm again [8]. On the other hand, cells with intensified resistance mechanisms (higher enzyme production or increased efflux pumps activity) appear much more frequently in the outer layers of the biofilm [6]. The microenvironment within biofilm not only hinders the penetration of antibiotics but also may favor the interspecific exchange of extrachromosomal DNA and spreading of acquired antibiotic resistance [9].

The formed biofilm on biotic surfaces (e.g., teeth) and abiotic surfaces (dental and orthopedic implants) is considered one of the most dangerous virulence factors [10]. It is a source of both acute and chronic infections and a cause of multi-million financial losses. This is especially true for biofilm-related infections of implants [11]. Therefore, there is a strong need to search for new biomaterials that will become less susceptible to biological contamination and will prevent the formation of biofilm that is often impossible to eradicate [3]. In ancient times, Hippocrates said “*Morbum evitare quam curare facilius est*”, which means that prevention is better than cure. This sentence seems to be relevant more than ever before.

The use of advanced bionanotechnology to synthesize new nanoapatites doped with noble metals is an extremely useful aspect of research limiting the spread of both drug resistance and biofilm formation. This is of paramount importance for clinical strains of bacteria with a high tendency to obtain drug-resistant genes and produce biofilms on implants and catheters such as MRSA (methicillin-resistant *S. aureus*) and VRE (vancomycin-resistant *Enterococcus*). Their importance was also emphasized by the World Health Organization by including them on its latest list as strains requiring the most intensive microbiological research [12]. All the above necessitates the search for a novel nanoapatite formulation with practical application in medical implantology.

Nanomaterials, and in particular nanoparticles (NPs), show wide antimicrobial activity against both Gram-positive (*Enterococcus*, *Staphylococcus*, and *Streptococcus*) and Gram-negative (*Escherichia*, *Pseudomonas*, etc.) bacteria. In this respect, however, many studies have shown a greater activity of NPs against Gram-positive bacteria than against Gram-negative ones [13,14]. This might be associated with differences in the cell wall structure. The presence of the outer membrane densely covered by lipopolysaccharide (LPS) in Gram-negative bacteria hinders the penetration of ions and NPs into their cells. In Gram-positive bacteria, the cell wall is not covered by the outer membrane and the peptidoglycan has numerous channels that allow the antimicrobial compounds to penetrate the interior [15]. It is also important that Gram-positive bacteria, such as *S. aureus* or *E. faecalis*, have a stronger negative surface charge than Gram-negative bacteria, which may also increase interactions with NPs [16].

Nanocrystalline hydroxyapatite (nHAp) constitutes the major inorganic component of bones and teeth [17]. It is characterized by high biocompatibility, bioactivity, and osteoconductivity. Therefore, in bone implantology, the use of synthetic apatite is of great interest, which can be designed to exhibit antibacterial activity when doped with noble metals such as silver (Ag), gold (Au), or palladium (Pd). The noble metals are recognized as being highly biocompatible due to their extreme chemical inertness and corrosion resistance in biological environments. Since many bacteria have acquired resistance to antibiotics (including multi-drug resistance, MDR), noble metals are increasingly used as bactericides [18]. Noble metal nanoparticles, due to their homogeneity and stability, are of great interest as one of the components of biomaterials. Their antibacterial activity is often multifactorial, which makes them highly effective. The mechanism of action, depending on the structure, can be varied and aimed at different targets in bacterial cells. Direct contact of bacteria with nanoparticles can lead to cell penetration; disruption of the cell membrane, protein alkylation, strong oxidative stress, and consequently, damage to the genetic material and death [17,19].

Bimetallic nanoparticles have a great potential for future biomedical applications, mainly due to the expected synergistic antibacterial effect [20,21]. This is reflected in several reports inducting the emerging resistance of bacteria to monometallic nanoparticles, in particular to the most commonly used as AgNPs [22,23,24]. However, many studies have shown that the main toxic element for bacteria is Ag, followed by Au, Pd, and thus, the best method seems to be to dilute silver with gold, palladium, or gold with palladium [25,26]. Following this thought, trimetallic nanoparticles could also improve antimicrobial efficacy.

The present research is aimed at determining the antiadhesive and antibiofilm properties of the newly synthesized bionanomaterials refined with noble metals against selected drug-resistant clinical strains of *Staphylococcus aureus* and *Enterococcus faecalis*. Detailed comparative studies of Ag-, Au-, or Pd-doped apatites with co-doped and triple-doped apatites in the form of pellets were carried out. Furthermore, the aim was to determine the cytotoxicity of noble metal-doped nanohydroxyapatites against eukaryotic cells. To the best of our knowledge, the present report describes for the first time the use of Ag^+^-Au^+^-Pd^2+^ ions-doped nanoapatites against bacteria.

## 2. Results

### 2.1. Physicochemical Properties of Obtained Apatites

To determine the crystallinity and phase structure, the samples of apatites were characterized by X-ray powder diffraction technique (XRPD). The diffraction patterns collected in Figure 1 show wide well-developed peaks confirming the obtained crystallized materials with nanometric sizes, which is in line with our previous research [27]. When compared with the ICSD (Inorganic Crystal Structure Database) data (Ca_10_(PO_4_)_6_(OH)_2_, ICSD-26204 [28]; Ca_10_(PO_4_)_6_Cl_2_, ICSD-24237 [29]), the phase of hydroxyapatite (nHAp) and chlorapatite (hereafter nClAp) have been identified. The presence of the mixed apatite phase (OH-Cl-Ap) when co-doping with Au^+^ and Pd^2+^ ions results from the use of chloride reagents during the synthesis. The chloride anion (Cl^−^) incorporates with high affinity into the hydroxyapatite matrix, which was described in our earlier paper [27]. In the case of monometallic doping (Figure 1A1), nHAp and nHAp: 1 mol% Ag^+^ are monophasic. The samples doped with 2 mol% Ag^+^ and 1 mol% or 2 mol% Au^+^ contain an additional phase of Ag^0^ (ICSD-22434 [30]) and Au^0^ (ICSD-52249 [31]), respectively. The additional peak at 2*θ* of about 38° is clearly visible, especially for Au^+^-doped apatites, identified as gold precipitates. Although the metal palladium peak is indistinguishable in the diffractograms of OH-Cl-Ap: 1 mol% or 2 mol% Pd^2+^ (probably overlaps with the apatite peak at about 2*θ* equal to 40° [32] (ICSD—257582 [33])), subsequent studies (UV–Vis absorption and SEM–EDS mapping) will show that Pd^0^ it is also present as an additional phase. On the XRPD diffraction patterns of double-doped apatites (Figure 1B1), extra peaks specific to metallic precipitates are observed for all materials. Similarly, for the triple-doped apatites (Figure 1C1), additional peaks of the metallic phase are present. Moreover, the arrangement of the most characteristic peaks in the 2*θ* range of 30–33° for these materials and for samples doped with 1 mol% or 2 mol% of Au^+^-Pd^2+^ (Figure 1B1) suggests that the predominant apatite phase for them is chlorapatite. In order to confirm this hypothesis, FTIR method was further applied as well as quantitative measurements of elements in the obtained samples using the SEM–EDS technique.

The FTIR spectra of the nHAp, nHAp doped with Ag^+^, as well as OH-Cl-Ap doped and co-doped with Ag^+^, Au^+^, and Pd^2+^ are shown in Figure 1A2–C2. The characteristic for apatites PO_4_^3−^ absorption bands are detected at about 472 cm^−1^ (doubly degenerate δ_2_ bending), 560 cm^−1^ and 600 cm^−1^ (triply degenerate δ_4_ bending), 964 cm^−1^ (symmetric non-degenerate stretching v_1_ vibrations), and 1022 cm^−1^ and 1090 cm^−1^ (asymmetric triply degenerate stretching v_3_ vibrations). The peaks belonging to the librational (ν_L_) and stretching (ν_S_) vibration of OH^−^ groups are localized at 631 cm^−1^ and 3574 cm^−1^, respectively. These bands are typical for hydroxyapatite [34]. However, for samples with the predominant chlorapatite phase, we do not observe or only slightly observe these bands, which can be seen in the presented spectra (Figure 1B2,C2, marked area) [27].

The EDS spectra are presented in the Appendix A, and the concentration of elements in apatite are gathered in Table 1. As found in the XRPD studies, chloride (Cl^−^) ions were successfully incorporated into the apatite structure of samples doped or co-doped with Au^+^ and Pd^2+^ ions. The more dopant, the greater Cl^−^ content (see Table 1). The chlorapatite phase is dominant for samples with 1 mol% or 2 mol% of Au^+^ and Pd^2+^ as well as for triple-doped apatites, which is in agreement with the XRPD results. Moreover, the ratio of cat./P (where cat. means Ca^2+^ + (Ag^+^/Au^+^/Pd^2+^)) for all apatites are very close (1.62–1.71) to the stoichiometric value (1.67), which confirms obtainment of the apatite structure for all synthesized materials (Table 1).

The obtained apatites were further analyzed with UV–Vis spectra (Figure 2) in order to detect characteristic surface plasmon resonance peak (SPR) for metallic nanoparticles. The spectra clearly indicated the formation of AuNPs (peaks at about 530–540 nm) for the apatite doped with Au^+^ as well double-doped with Au^+^ and Ag^+^ [35]. There is a slight boost in the nHAp: 2 mol% Ag^+^ spectrum—a broad band with a maximum of approximately 423 nm characteristic for AgNPs [36]. Interestingly, the addition of palladium caused a lack of SPR of PdNPs in the UV–Vis spectra. Moreover, for the Pd-apatites co-doped with Au^+^ and/or Ag^+^, the plasmon bands of AgNPs and AuNPs were also not observed in the spectra, similar to PdNPs. Sivamaruthi et al. [36] described this phenomenon as an encapsulation with PdNPs, which is thick enough to suppress the SPR peak from AgNPs or AuNPs.

Further, to confirm the presence of metallic nanoparticles, especially Pd^0^, it was decided to prepare SEM images together with element mapping (SEM–EDS). The images are presented in Appendix A and Figure 3. Metallic precipitations, especially of Au^0^, are clearly visible as brighter spots in the SEM images (marked with orange arrows on the element maps). Moreover, SEM images at 5000× magnification indicate that the pellets are not clearly smooth, but rather rough. This observation correlates with our previous research proving that roughness is one of the most important features regarding bacterial adhesion and cell attachment ability to the surface [34].

### 2.2. Microbiological Analysis of Nanoapatites

The main purpose of the present research was evaluation of the potential use of nanoapatites in biomedical applications against bacterial biofilms. The microbiological part of the investigation was performed in three stages described below:At the beginning, our purpose was to select clinical drug-resistant strains of *Enterococcus faecalis* and *Staphylococcus aureus* with the strongest intensity of biofilm production. Among tested strains, *E. faecalis* VRE 200 and *S. aureus* MRSA P19 were selected and used in the next stages of our research.Futher research was focused on evaluation of basic antimicrobial properties (MIC, MBC, and FICI values) as well as antibiofilm properties of nanoparticles against these strains. Experiments testing the release of ions from nanoparticles via the ICP-OES method were also perfomed. The antibiofilm assays included qualitative and quantitative analysis of the impact of nanoparticles on bacterial cell adhesion and biofilm formation. These measurements were performed using fluorescence microscopy and SEM (scanning electron microscopy) together with computational analysis of the obtained pictures.The last stage was focused on measurements of potential toxicy of nanoparticles. For this purpose, MTT assays, adhesion of Balb/3T3 mouse embryonic fibroblasts on nanoparticles, as well as influence of bacterial biofilm on the viability of fibroblast cells on nanoapatites were performed. Fluorescence microscopy and computational analysis of images were used to obtain the results.

Detailed results of each experiment are described in the next paragraphs.

#### 2.2.1. Selection of Bacterial Strains Based on Their Biofilm Production

The selection of bacterial strains in the very first stages of our research was performed using the colorimetric and fluorescence-based measurement of biofilm production. The results are presented in Figure 4. Among all tested drug-resistant strains, the strongest production of biofilm was exhibited by *S. aureus* MRSA P19. In the case of enterococci, the best result was obtained for *E. faecalis* VRE 200. Based on the obtained data, these two strains were used in the next stage of our research.

#### 2.2.2. Evaluation of MIC and MBC of Nanoapatites against Selected Strains

The results of the evaluation of MIC and MBC of tested biomaterials against *E. faecalis* VRE 200 and *S. aureus* MRSA P19 are presented in Table 2. The lowest MIC was exhibited for the triple-metallic component of 2Ag-2Au-2Pd nanoapatites against both strains (128 µg/mL). Moreover, MIC and MBC showed the same values, which may be interpreted as domination of bactericidal effect over bacteriostatic one. Similar activity was also observed for triple-component 1Ag-1Au-1Pd nanoapatites. The results seem to be variable and show that sometimes formulations were more active against *S. aureus* MRSA P19 (MIC for 2Ag and 2Ag-2Au equal to 512 µg/mL and 256 µg/mL, respectively), while other times more effective against *E. faecalis* VRE 200 (1Au, 2Au, 1Pd, 2Pd, and 1Au-1Pd (for all MIC = 4096 µg/mL) and 1Ag-1Au-1Pd with MIC being 128 µg/mL). Generally, in the case of mono-metallic nanoapatites, MIC and MBC values were usually much higher than those observed for double- and triple-component nanoapatites and reached values ranging from 1024 µg/mL to above 8192 µg/mL.

In summary, most of the one-component nanoapatites exhibit bacteriostatic properties, while some selected two-component nanoapatites and all three-component nanoapatites are bactericidal also. These formulations show slightly stronger bactericidal activities against *E. faecalis* VRE 200 than against *S. aureus* MRSA P19.

#### 2.2.3. Evaluation of Fractional Inhibitory Concentration Index of Nanoapatites Composition

In total, there were six, three, and seven combinations showing synergism, additivity, and neutral effect, respectively. It is important to highlight that in case of both tested bacterial strains (VRE 200 as well as MRSA P19) triple-component metallic nanoparticles exhibited antimicrobial synergism (FICI = 0.12–0.31). In the case of double-component metallic nanoparticles, all types of interactions except antagonism were observed. In that respect, synergism was observed for 1Ag-1Pd against VRE 200 and for 2Ag-2Au against MRSA 19. In summary, a clear dependency between the nanoparticles’ compositions and antibacterial effects was difficult to be made. However, the fact that triple-component metallic nanoparticles presented synergism in all cases suggests that noble metals within these formulations may differ in their target sites and mechanisms of action. Detailed results evaluating FICI of nanoapatites are presented in Figure 5A.

#### 2.2.4. Release of Metal Ions in the TSB Medium by (ICP-OES)

The results of the release of metal ions (Ag^+^, Au^+^, and Pd^2+^) from the tested biomaterials at various incubation times show a significant increase in the amount of Ag^+^ ions with silver-doped and co-doped nanoapatites (Figure 6B). In particular, biomaterial 1Ag (1) and 2Ag (2) released nearly 30 µg/mL and 63 µg/mL of Ag^+^ ions, respectively. Other metallic elements were only detected in trace amounts. This suggests that the silver released into the medium may increase the antimicrobial effect (Figure 5B).

#### 2.2.5. Influence of the Studied Nanoapatites on the Adhesion and Biofilm Formation by Drug-Resistant *E. faecalis* VRE 200

The anti-adhesive and anti-biofilm activity against *E. faecalis* VRE 200 of the tested nanoapatites for three different incubation periods (12 h; 24 h; and 48 h) proved to be differentiated. After 12 h, most of the tested biomaterials inhibited adhesion and biofilm formation by the tested strain in comparison with the control, non-doped nanohydroxyapatite (52% of the surface area). The exceptions were the gold-doped 1Au (3) and 2Au (4) samples which did not inhibit biofilm formation and were similar to the control (Figure 6A).

After 24 h of incubation, the biofilm area in the control was similar to that detected after 12 h and was at the level of 51%. In addition, a similar biofilm area was obtained for 1Au (3) being 41% and for 2Au (4) being 42%. A slight increase in the biofilm area to 5.5% for 1Ag-1Au (7) and 2% for 2Au-2Pd (12) was also observed (Figure 6A).

After 48 h, the biofilm area reached 72% in the control. For 1Au (3) sample, the biofilm area did not change (42%). In case of 2Au (4) and 1Au-1Pd (11) samples, the biofilm area decreased and increased to 15%, respectively (Figure 6A).

The presented results indicate that almost all tested nanoapatites caused strong inhibition of adhesion and biofilm formation of *E. faecalis* VRE 200. The best anti-adhesive and anti-biofilm abilities were exerted by the following nanoapatites: 1Ag (1); 2Ag (2); 1Pd (5); 2Pd (6); 2Ag-2Au (8); 1Ag-1Pd (9); 2Ag-2Pd (10); and 1Ag-1Au-1Pd (13) samples. For further information see Figure 7.

#### 2.2.6. Influence of the Studied Nanoapatites on the Adhesion and Biofilm Formation by Drug-Resistant *S. aureus* MRSA P19

The anti-adhesive and anti-biofilm activities of the nanoapatites against *S. aureus* MRSA P19 tested for three different incubation periods (12 h; 24 h; and 48 h) were determined in a similar manner as for enterococci. The surface area by the adhered bacterial cells reached 58% in the control. It was observed that all the tested nanoapatites caused the strong inhibition of MRSA P19 adhesion compared with the control. The lowest level of adhesion was observed for samples 1Au (3) and 2Au (4), where adhered bacteria covered 29% and 8% of the sample area, respectively (Figure 6B).

After 24 h, the biofilm area reached 86% in the control. An increase in biofilm area was also observed on a few nanoapatites. For the apatite 1Au (3) sample, a surface area of 78% was achieved, while for 2Au (4) sample it reached only 8%. For some double-component nanoapatites being 1Ag-1Au (7), 1Ag-1Pd (9), and 2Au-2Pd (12) samples, the biofilm area reached 9%, 2%, and 1%, respectively. Interestingly, biofilm appeared for the first time on triple-doped apatites reaching 10% and nearly 12% for 1Ag-1Au-1Pd (13) and 2Ag-2Au-2Pd (14) samples, respectively (Figure 6B).

After 48 h of incubation, the biofilm area on the nanohydroxyapatite control reached 85%. In sample 1Au (3), the biofilm area was similar to the control reaching 81%, while sample 2Au (4) was equal to 17%. For the nanoapatite 1Ag-1Au (7) sample, the biofilm covered 8% of the surface area, while the 1Ag-1Au-1Pd (13) and 2Ag-2Au-2Pd (14) samples reached 11% and 12%, respectively. It is worth mentioning that biofilm was observed for the first time for 1Ag-1Pd (9) and 2Au-2Pd (12) samples, but it was neglected and accounted for 1–2% (Figure 6B).

In summary, the results were similar to that obtained for *E. faecalis* VRE 200, although it was noticed *S. aureus* MRSA P19 produced more biofilm both in the control and some of the nanoapatites tested. The following nanoapatites showed the strongest antiadhesive and antibiofilm properties: 1Ag (1); 2Ag (2); 1Pd (5); 2Pd (6); 2Ag-2Au (8); 1Ag-1Pd (10); and 1Au-1Pd (11) samples. For further information see Figure 8.

#### 2.2.7. Cytotoxicity of the Studied Biomaterials on Balb/3T3 Fibroblasts (MTT)

The results of the cytotoxic activity of the tested nanoapatites against the Balb/3T3 fibroblasts cell line showed only a slight decrease in the fibroblasts’ viability for most biomaterials (5–10% reduction compared with the control). Only for the triple-doped nanoapatites was a more significant decrease in the viability of fibroblasts observed; this is up to 80% for the 1Ag-1Au-1Pd (13) and 2Ag-2Au-2Pd (14) samples (Figure 6C).

#### 2.2.8. Adhesion of Balb/3T3 Fibroblasts to Surface Tested Nanoapatites

Experiments determining adhesion of Balb/3T3 fibroblasts to the surface of the tested biomaterials showed that fibroblasts were able to adhere to up to 12% of the nanohydroxyapatite surface (the control sample) after 24 h. For all single-doped nanoapatites (1–6) and double-doped 1Ag-1Au (7), 1Au-1Pd (11), and 2Au-2Pd (12) samples, an insignificant decrease in adhesion was observed. For some double-doped nanoapatites, 1Ag-1Pd (9) and 2Ag-2Pd (10) samples, and triple-doped nanoapatites, 1Ag-1Au-1Pd (13) and 2Ag-2Au-2Pd (14) samples, a decrease in adhesion to the biomaterials of 4–6% was observed. The highest decrease in the adhesion level was observed for the 2Ag-2Au (8) sample, which was only about 2% (Figure 6D). For further information see the Appendix A.

#### 2.2.9. Influence of the Biofilm Produced by *E. faecalis* VRE 200 on the Viability of Balb/3T3 Fibroblasts

The assays estimating an impact of *E. faecalis* VRE 200 and its biofilm on the viability of Balb/3T3 fibroblasts adhered on the tested biomaterials showed that almost all tested formulations increased the viability of eukaryotic cells in comparison with the control. The pure nanohydroxyapatite, being a control, provided the viability of only 10% of fibroblasts and was a result of intensive biofilm overgrowth (78%). Interestingly, a strong stimulation of the biofilm formation by nanoapatite 1Au (3) sample was observed (99% of the nanoapatite surface), although it did not cause such a strong decrease in the fibroblasts’ viability as in the control (48%). The strongest inhibition of bacterial adhesion and biofilm formation, while maintaining the high fibroblast cell viability, was achieved for 2Pd (6), 1Ag-1Au (7), 1Ag-1Pd (9), and 2Ag-2Pd (10) samples and ranged between 59–69%. The strongest inhibition of both bacterial adhesion and the fibroblast viability was demonstrated for 2Ag-2Au (8) and 2Ag-2Au-2Pd (14) samples (Figure 6E). For further information see the Appendix A.

#### 2.2.10. Influence of the Biofilm Produced by *S. aureus* MRSA P19 on the Viability of Balb/3T3 Fibroblasts

Similarly as above, the results concerning the influence of the *S. aureus* MRSA P19 strain and its biofilm on the viability of Balb/3T3 fibroblasts adhered on the tested biomaterials showed protective activity of nanocomposites on eukaryotic cells. This effect was visible even when the bacterial biofilm was not so strongly inhibited. The pure nanohydroxyapatite for which the biofilm after 24 h reached 71% and the viability of fibroblasts counted for 2% was used as control. The most active biomaterial was the 1Ag (1) sample which almost completely inhibited staphylococcal biofilm and increased the viability of fibroblasts to almost 100%. The nanoapatites having a protective effect on the fibroblasts’ viability (28–43%) but not inhibiting biofilm formation were: 2Au (4), 2Ag-2Pd (10), 2Au-2Pd (12), and 1Ag-1Au-1Pd (13) samples (Figure 6F). For further details see the Appendix A.

#### 2.2.11. Viability of *E. faecalis* VRE 200 Biofilm on the Surface of Nanoapatites

The results of the viability of *E. faecalis* VRE 200 grown as biofilm on the surface of the tested biomaterials showed that the tested nanoapatites have not shown bactericidal effect but rather a bacteriostatic one. In general, after 12 h of incubation no decrease in the biofilm viability was observed. Only 2Au (4) (25% of dead cells (DC)), 2Pd (6) (11% of DC), and 2Au-2Pd (12) (66% of DC) had ability to decrease in the biofilm viability. After 24 h, an increase in biofilm was observed on the single-doped nanoapatites, which was predominantly characterized by the high viability. A larger decrease in the bacterial viability was presented for the double- and triple-doped nanoapatites: 1Ag-1Au (7) (22% DC), 1Au-1Pd (11) (44% DC), 2Au-2Pd (12) (42% DC), 1Ag-1Au-1Pd (13) (83% DC), and 2Ag-2Au–2Pd (14) (88% DC) samples. After 48 h of incubation, a slight decrease in the biofilm viability was observed on the single-doped nanoapatites and a strong decline was detected for the double-doped 2Au-2Pd (12) (59% DC) and three-doped 2Ag-2Au-2Pd (14) (58% DC) apatites. It is noteworthy that a reduced amount of biofilm was observed after 48 h on the surface of nanoapatites (Figure 7). In summary, the tested nanoapatites presented usually bacteriostatic activity, strongly inhibiting both adhesion and biofilm formation of *E. faecalis* VRE 200. Only a few among the double-doped and triple-doped nanoapatites showed bactericidal activity. For the single-doped nanoapatites, slow bacterial adaptation mechanisms can be observed; however, for the triple-doped ones, the amount of biofilm decreased after 48 h, suggesting that triple-component formulations may reduce the risk of adaptive changes.

#### 2.2.12. Viability of Biofilm *S. aureus* MRSA P19 on the Surface of Nanoapatites

Again, the viability of *S. aureus* MRSA P19 cells in biofilm on the surface of the tested biomaterials showed a strong consistency with the results obtained for *E. faecalis* VRE 200. After 12 h, mostly no biofilm was observed for the double-doped nanoapatites. Reduced cell viability of biofilm was observed for the following nanoapatites: 2Au (4) (25% of DC), 2Au-2Pd (12) (73% of DC), 1Ag-1Au-1Pd (13) (89% of DC), and 2Ag-2Au-2Pd (14) (82% of DC) samples. After 24 h, an increase in the biofilm viability was observed for most of the tested biomaterials. Only for 2Au-2Pd (12) (11% of DC) and the triple-doped 1Ag-1Au-1Pd (13) (28% of DC) and 2Ag-2Au-2Pd (14) (30% of DC) samples was higher bactericidal activity observed. After 48 h, an increased cell death was noticed for biofilms created on the surface of the following nanoapatites: 1Ag-1Au (7) (11% of DC), 1Ag-1Pd (9) (12% of DC), and triple-doped 1Ag-1Au-1Pd (13) (22% of DC) and 2Ag-2Au-2Pd (14) (32% of DC) samples (Figure 8). In conclusion, all double-doped nanoapatites were most effective in inhibiting bacterial adhesion and biofilm formation (except the Au^+^ ions dopant sample). Triple-doped nanoapatites had the strongest anti-biofilm properties against *S. aureus* MRSA P19.

#### 2.2.13. Adhesion and Biofilm Formation of Drug-Resistant Clinical Strains of *E. faecalis* VRE 200 and *S. aureus* MRSA P19 Observed by Scanning Electron Microscopy (SEM)

Based on the obtained results, nanoapatites with a double amount of nanoparticles (2X, 2X:2Y, and 2X:2Y:2Z) were selected for scanning electron microscopy imaging (Figure 9) as they showed stronger antimicrobial properties than nanoapatites with a single amount of nanoparticles (1X, 1X:1Y, and 1X:1Y:1Z).

For *E. faecalis* VRE 200, evenly adhered bacterial cells with elements of more complex spatially organized biofilms were imaged in the control. The surface of the 2Au (4) sample was similar to the control, but it also can be observed that cells of *E. faecalis* VRE 200 tend to communicate with each other and the surface by adhesive fimbriae. The surfaces for nanoapatites with 2Ag (2), 2Pd (6), 2Ag-2Au (8), 2Ag-2Pd (10), and 2Au-2Pd (12) samples were highly abundant with elements of fine cusps (presumed nanoparticles) and presented some cellular debris with no intact bacteria observed. Similarly, for triple-doped nanoapatite (14), a lack of bacteria was observed and only some elements of cellular debris were spatially distributed.

For *S. aureus* MRSA P19, the biofilm on the control surface was abundant and well-developed. Again, on the surface of the 2Au (4) biomaterial, numerous but small clusters of bacterial cells arranged in vertical spatial structures were observed. For nanoapatite 2Ag (2) and 2 Pd (6), a lack of bacterial cells was detected. In these cases, structures resembling narrow flakes or bands were noticed. Similarly, no bacteria were visualized on double-component samples 2Ag-2Au (8) and 2Ag-2Pd (10). For the 2Au-2Pd (12) and 2Ag-2Au-2Pd (14) samples, single non-aggregated cells of *S. aureus* MRSA P19 were observed.

In summary, it was proved that the studied nanoapatites can effectively reduce biofilm formation of both tested bacterial strains. Some nanoapatites, such as the 2Au (4) sample, can strongly influence the production of pilus-like structures by bacteria.

## 3. Discussion

The growing resistance of bacteria and their ability to form biofilms are both important problems in the healthcare system, posing a threat to the life of many patients worldwide [37]. Thus, the search for novel nano-biomaterials, which can be used in medicine, is no longer a need but a necessity. As nanoapatites doped with noble metals possess promising antimicrobial properties, research attempts to produce bionanomaterials with antibacterial and antibiofilm action against drug-resistant clinical strains of *E. faecalis* VRE and *S. aureus* MRSA have been made. We decided to use three different noble metals (Ag, Pd, and Au; all in various combinations) because a lot of encouraging reports on their antibacterial properties and low cytotoxicity at low concentrations have been published.

The first component of our nanoformulation was silver. This metal occupies a leading position in the area of coating the surface of medical implants [17,38,39]. In the analyzed model of nanoapatites, it is particularly important that the silver nanoparticles (AgNPs) may destroy bacterial cells by direct contact [40]. Another mechanism involves the extracellular and intracellular generation of silver-stimulated reactive oxygen species (ROS) through a catalytic effect in the Fenton reaction. Moreover, the released silver ions could enhance the antibacterial effects we observed. The main one is the release of Ag ions taken up by the cells, which then disrupts ATP production and DNA replication. Increased amounts of released silver ions into the medium (only for single Ag^+^-doped nanoapatites) may enhance the antimicrobial effect [41]. Moreover, AgNPs, when combined with antibiotics, can synergistically fight bacteria, including drug-resistant ones, such as, e.g., *Salmonella* Typhimurium. In this case, it was presented that Ag^+^ release from AgNPs is a factor increasing the antibacterial efficacy of the AgNPs-antibiotic complex [42]. Although AgNPs are widely used against bacteria, it should be mentioned that their further applications may be limited due to their cytotoxicity against human cells. Therefore, modifications of AgNPs are used more and more often in order to reduce this harmful parameter. In this context, it was shown that the alloy nanostructure of gold nanoparticles inlaid on AgNPs and synthesized using egg white proteins (Au-AgNPs) presented an enhanced antimicrobial activity and reduced Ag cytotoxicity [43]. A similar mechanism of reducing the silver content in the component while maintaining its microbial activity was shown by Qing et al. [44], which placed Au-AgNPs into chitosan as wound dressings.

The second component used in the current research was gold. Next to silver, AuNPs are also widely applied in biomedical materials. Overall, both Au (I) and Au (III) ions are highly toxic to bacteria, however, the mechanism of the antimicrobial action of gold nanoparticles (AuNPs) is still poorly understood [35,45,46]. It has been observed that when the size of AuNPs drops drastically (1–2 nm), the mechanism of action of gold may be completely different from that of silver. In that respect, the intracellular ROS production is of minor importance in terms of antimicrobial toxicity, but the size effect is critical leading to the irreversible binding to biopolymers [47]. Some researchers postulate that the observed antibacterial effect is the result of co-existing chemical states of gold with AuNPs [45,46,48]. Our research has shown that single-gold ion-doped nanoapatites exhibit the lowest antimicrobial activity out of all tested nanosized formulations which does not correspond to the literature [36]. Moreover, AuNPs, in a similar manner as AgNPs, can be combined with other antimicrobial compounds (e.g., antibiotics) and synergistically inhibit the growth of bacteria, especially antibiotic-resistant ones [49].

The third component of metal-doped nanoapatites was palladium. This metal is also known for its antibacterial activity, although it is not so widely investigated compared with the two previous ones [32,36,50].

Interestingly, our research has shown that single-gold-doped nanoapatites exhibit the lowest antimicrobial activity out of all tested nanoformulations, which does not correspond to the literature [36]. Its mechanism of action, however, is not well understood [51,52]. Antibacterial activity of Pd nanoparticles (PdNPs) via generation of ROS has been associated with their membrane-penetration capacity. Pd octahedrons showed stronger penetration of membranes of Gram-negative bacteria than Pd nanocubes, while an inverse correlation was presented for Gram-positive bacteria [53]. Other studies suggest that Gram-positive bacteria (*S. aureus* and *Bacillus subtilis*) may be less sensitive to the action of nanoparticles incorporated on the surface of biomaterials compared with Gram-negative bacteria (*E. coli* and *Salmonella*). This seems to be strongly associated with differences in the thickness of the cell wall between these two bacterial groups [41,52,54]. Our checkerboard assays highlighted the presence of synergism between components of the following compositions: 1Ag-1Pd (9) (against VRE 200) and both 1Ag-1Au-1Pd (13) and 2Ag-2Au-2Pd (14) (against VRE 200 and MRSA P19). This suggests that the mechanism of action of the metal component within our nanoformulations is different and that noble metals when used together may potentiate their antimicrobial properties. The results also perfectly correspond to the bacteriostatic and bactericidal effect of the tested nanoapatites [55,56]. Moreover, Zhang et al. [57] showed that the antibacterial activity of compounds is influenced not only by the type of doped metal, but also by the structure of the compound (e.g., the number of ligands forming it). In line with this, it was noticed by others that two ligands or two different types of ligands show better antimicrobial activity, especially against *B. subtilis* and *S. aureus*, than a single ligand [58,59]. Moreover, the nature of the ligand is also very important, e.g., when the ligand has a chelating effect it will increase the antimicrobial activity of the compound [57]. Chelation reduces the polarity of the metal atom and promotes the penetration of the central atom through the lipid layer of microbial cell membranes by increasing its lipophilic character [55,56,57,60]. In turn, studies by Sobierajska et al., conducted on doped and co-doped (Li^+^ and Eu^3+^) nanoapatites, showed mostly no antibacterial activity. Here, only co-doped nanoapatite 1 mol% Eu^3+^/2 mol% Li^+^ ions were shown to be active against *P. aeruginosa* [27].

In addition to the antibacterial properties of the tested nanoapatites, we have observed that our nanoformulations can effectively reduce biofilm formation and change the spatial structure formation of both VRE 200 and MRSA P19. The one mechanism explaining this phenomenon may be strictly related to the antibacterial action of the tested nanoapatites. This is consistent with many studies showing such activity [41,52,61]. Although, it is important to highlight that most of the research that investigated the viability of biofilms treatment with nanoformulations used only one-doped metal component biomaterials, such as gold [52,61,62] or silver [63,64]. This was mostly a result of disruption of microbial cell membranes/cell wall and the oxidative stress-dependent inference with metabolism [40,65,66]. Alternatively, the anti-biofilm activity of the studied nanoapatites may be the result of interactions between bacteria and NPs, and formation of anti-adhesive surfaces preventing biofilm development. This suggests that direct surface interactions predominate on the surface of metal-ions-doped nanoapatites because single-ion-doped (Ag^+^ and Pd^2+^ ions) and double-ions-doped nanoapatites (especially Ag^+^-Pd^2+^ or Ag^+^-Au^+^ pair ions) showed a very strong antiadhesive activity. Naidi et al. [67] also showed that metal-ions-doped compounds (Zr/Sn-dual doped CeO_2_NPs) can increase their antibiofilm activities against *S. aureus* and *Listeria monocytogenes*. In turn, Thakur et al. [68] investigated the effect of co-doping (Ag^+^ and Zn^2+^) on the bactericidal properties of CuO nanoparticles and found that the growth inhibition of *E. coli*, *S. aureus*, and *P. aeruginosa* is higher for co-doped CuONPs than those with pure CuONPs.

Research on composite materials doped with noble metals used for dental implants similarly showed a higher antibiofilm activity for silver-encrusted composites than for gold incorporated ones [69]. Moreover, in the current article, it was noticed that the single-doped nanoapatites may promote slow but visible bacterial adaptation mechanisms visualized by the intensified biofilm formation, although for the double- and triple-doped nanoapatites, the amount of biofilm decreased significantly. Additionally, we have detected that some nanoapatites, such as these with 2Au (4), can strongly influence the production of pilus-like structures by bacteria. This highlights another very interesting mechanism of adaptation to severe conditions on the surface of metal-doped nanoapatites by providing an increase in the surface-bacteria distance and lower antimicrobial activity [70].

The determination of side effects of the tested biomaterials against a human cell line was the last part of the current study. Our investigation of this aspect included the establishment of cytotoxicity of the studied biomaterials on Balb/3T3 fibroblasts, adhesion of fibroblasts to the surface of nanoapatites, as well as influence of the biofilm produced by VRE 200 and MRSA P19 on the viability of fibroblasts. We chose to work with fibroblasts as this type of cell plays a very important role in the regeneration of bone defects (they adhere as one of the first) and in the acceptance of implant biomaterials, including artificial bones [71]. Our results suggest that most of the tested nanoapatites, regardless of the doped metals used, were non-toxic to fibroblasts. This is consistent with reports of others who also did not observe a decrease in the viability of fibroblasts exposed to biomaterials doped with Ag and Pt [40,72]. Only in the case of triple-doped metal apatites di d we notice a minor decline in the survivability of fibroblasts (up to 80%) and a decrease in the adhesion of these cells to the surface of nanoapatites. However, this requires further research as it may only be related to the longer adaptation of fibroblasts over time to the surface of nanoapatites, as suggested by Keivani et al. in studying the proliferation time of fibroblasts on modified nanoapatites [73]. It is also important to note that most of the studied biomaterials increased the viability of fibroblasts in the presence of biofilms produced by the tested strains (especially visible for MRSA P19). This means that even in the case of contamination of implants by streptococci or enterococci, metal-doped nanoapatites can provide advantages to fibroblasts and contribute to faster body regeneration [74].

## 4. Materials and Methods

### 4.1. Synthesis of Nanoapatites

The apatites nanocrystals of (nHAp and OH–Cl–Ap) were synthesized by the co-precipitation method at the Institute of Low Temperature and Structure Research, Polish Academy of Sciences in Wroclaw, Poland. Analytical grades of Ca(NO_3_)_2_·4H_2_O (≥99% Acros Organics, Schwerte, Germany), (NH_4_)_2_HPO_4_ (≥98% Avantor Performance Materials Poland S.A, Gliwice, Poland), AgNO_3_ (≥99.9% Avantor Performance Materials Poland S.A, Gliwice, Poland), PdCl_2_ (anhydrous, 60% Pd basis, Sigma-Aldrich, Steinheim, Germany), Au (Mennica-Metale, Radzymin, Poland), HNO_3_ (65% Suprapur^®^, Supelco Analytical, Munich, Germany), and HCl (30%, Suprapur^®^, Supelco Analytical, Munich, Germany) were used as the starting reagents. The pH was set using NH_3_∙H_2_O (99%, Avantor Performance Materials Poland S.A., Gliwice, Poland). The concentration of Ag^+^, Au^+^, and Pd^2+^ ions was set at the level of 1 mol% and 2 mol% to the overall molar content of Ca^2+^ cations. Firstly, water-soluble HAuCl_4_ was obtained using aqua regia (a mixture of nitric acid and hydrochloric acid in a molar ratio of 1:3). Palladium nitrate was obtained from the reaction of PdCl_2_ with HNO_3_. All required substrates were dissolved and then mixed on the magnetic stirrer. The pH of the mixture was adjusted to 10 with an ammonia solution and mixed at 70 °C for 3 h. The obtained precipitated product was dried at 90 °C for 24 h and heat-treated at 450 °C for 3 h. A hydraulic press with a pressure of 5 kN was used to obtain pellets with a diameter of 6 mm.

### 4.2. Characterization of Obtained Materials

PANalytical X’Pert Pro X-ray diffractometer (Malvern Panalytical Ltd., Malvern, UK) with a Cu–Kα radiation at *2θ* range from 10° to 60° (exposure time of 2 h) was applied to determine structure and crystallinity. The obtained diffraction patterns were juxtaposed with standards from the Inorganic Crystal Structure Database (ICSD). Fourier-transform infrared spectroscopy (FTIR, Biorad 575C spectrophotometer, Hercules, CA, USA) in a frequency range of 4000–400 cm^−1^ was used to determine the functional groups of the obtained materials. This was performed using independent replicates. The UV–Vis spectra were recorded on an Agilent Cary 5000 UV–Vis-NIR spectrophotometer (Agilent Technologies, Santa Clara, CA, USA) with a spectral bandwidth of 1 nm in the range of 200 to 800 nm (50,000–12,500 cm^−1^). The morphology, element concentration, and mapping were performed using a scanning electron microscope (SEM, FEI Nova NanoSEM 230, Hillsboro, OR, USA) with an energy-dispersive X-ray spectrometer (EDS, Genesis XM4, Austin, TX, USA). The EDS spectra were recorded three times for each sample and the calculated value was an average result.

### 4.3. Microbiological Analysis of Nanoapatites

#### 4.3.1. Strains and Growth Conditions of Bacteria

In this study, we used the following drug-resistant bacterial strains (*Enterococcus faecalis* VRE V583; *E. faecalis* VRE 123; *E. faecalis* VRE 200; *E. faecalis* VRE 037, *Staphylococcus aureus* MRSA P19; *S. aureus* MRSA P5; *S. aureus* MRSA B825; and *S. aureus* MRSA P31). Bacterial strains were cultured in Tryptic Soy Broth BD (TSB). The strains were incubated aerobically for 24 h at 37 °C. Overnight, microorganism cultures were centrifuged, washed with PBS (pH = 7.4) and suspended in fresh TSB to obtain suitable optical density. The strains were identified by 16S rRNA analysis and are in the collection of the Department of Microbiology of the Wroclaw Medical University in Poland.

#### 4.3.2. Assessment of Biofilm Production Ability

The measurement was performed according to the modified method [10]. The biofilm on the microplate was stained using LIVE/DEAD BacLight Bacterial Viability Kit (Thermo Fisher Scientific, Waltham, QC, Canada). For further information see Section 4.3.11. below. The experiment was conducted with three independent repetitions.

#### 4.3.3. Minimal Inhibitory and Bactericidal Concentrations (MIC and MBC)

The minimal inhibitory concentrations (MIC) were determined according to the modified protocol described previously [10,75].

#### 4.3.4. Fractional Inhibitory Concentration Index (FICI)

The potential existence of positive interactions in the antibacterial activity (additive or synergistic) between the tested noble metals on nanoapatites was performed using a checkerboard assay. To evaluate this, bacterial suspensions (10^5^ CFU mL^−1^) were prepared in TSB medium and placed in 96-well microplates. Concentration gradients of each component tested were located within the *x*- and *y*-axes, respectively. The microplates were incubated for 24 h at 37 °C with shaking (400 RPM). Based on the obtained MIC values of nanoapatites doped with 1 metal ion in the previously characterized proportions (1 and 2), compounds (**1**–**6**) and nanoapatites co-doped with two and three metal ions in identical proportions ((1-1; 2-2; 1-1-1; 2-2-2) and compounds (**7**–**14**)) were calculated using a fractional inhibitory concentration index (FICI). The interactions were interpreted based on the calculation of FICI, in which: Σ FIC ≤ 0.5 synergism; 0.5 < Σ FIC ≤ 1 additivity; 0.5 < Σ FIC ≤ 1 neutral; and 1 < Σ FIC ≤ 4 antagonism interaction, respectively [76,77].

#### 4.3.5. Release of Metal Ions in the TSB Medium by Inductively Coupled Plasma Optical Emission Spectrometry (ICP-OES)

A bench-top optical emission spectrometer, model 720 (Agilent, Santa Clara, CA, USA), with an axially viewed Ar-ICP and a 5-channel peristaltic pump was used to measure the concentrations of trace elements. The instrument was equipped with high-resolution echelle-type polychromator and a VistaChip II CCD detector (Agilent) cooled down to −35 °C on a triple-stage Peltier device. The plasma was sustained in a standard 1-piece, low-flow, extended quartz torch with a 2.4 mm inside diameter injector tube. A single-pass glass cyclonic spray chamber and a OneNeb pneumatic concentric nebulizer made of a high-tech PFA and PEEK polymers were used to introduce the sample solutions by pneumatic nebulization. Operating conditions recommended by the manufacturer for solutions containing high levels of dissolved solids were applied: an RF power of 1200 W, a plasma gas flow rate of 15.0 L min^−1^, an auxiliary gas flow rate of 1.5 L min^−1^, a nebulizer gas flow rate of 0.75 L min^−1^, a sample flow rate of 0.75 mL min^−1^, a stabilization delay of 15 s, a sample uptake delay of 30 s, a rinse time of 10 s, a replicate read time of 1 s, and 3 replicates. A fitted background mode with 7 points per line profile was applied for the background correction. Background-corrected intensities of analytical lines were used for calibration graphs. Measurements of the ion release from the surface of the discs (5 mm in diameter and 3 mm in thickness) in the water phase (TSB medium) were measured for three incubation times: 12 h, 24 h, and 48 h at 37 °C with shaking (400 RPM). Two negative controls were used: (C-) TSB medium and C HAp as nanohydroxyapatite [78].

#### 4.3.6. Adhesion and Biofilm Formation of Bacterial Strains on Nanoapatites

Analysis of the adhesion and biofilm formation of two bacterial strains (*E. faecalis* VRE 200 and *S. aureus* MRSA P19) on nanoapatites was performed using fluorescence microscopy (FM). For further information see Section 4.3.11. below. Selected bacterial strains were added to TSB medium to obtain a final concentration of 10^5^ CFU mL^−1^. Sterile discs of tested nanoapatites doped with noble metals were added to 48-well non-adherent plates, while non-doped a nanohydroxyapatite was used as a control. A 1 mL amount of medium with the strain of bacteria was added to a microtitration plate. Cultures were performed for 12 h, 24 h, and 48 h at 37 °C with shaking (400 RPM). After the above-mentioned incubation periods, the discs were washed 3 times with sterile PBS (pH = 7.2) and prepared for staining with fluorescent probes. The experiment was conducted with three independent repetitions.

#### 4.3.7. Fibroblast Cell Culture

Balb/3T3 mouse embryonic fibroblasts were cultured in DMEM medium supplemented with 10% fetal bovine serum, 1 mM L-glutamine, and a 1% penicillin/streptomycin solution. The cells were cultured at 37 °C and 5% CO_2_ until the cell density of 10^6^ mL^−1^ was obtained. Such cells were then seeded in 24-well tissue microplates (2 × 10^4^ cells per well) and were allowed to adhere overnight under the conditions described above. Cytotoxicity tests were carried out on the cells prepared in this way.

#### 4.3.8. Cell Proliferation Assays (MTT Test)

Balb/3T3 mouse embryonic fibroblasts were incubated with the tested biomaterials for 24 h. To assess their viability, the MTT assay was used as previously described [79]. After incubating the cells with the tested hydroxyapatite discs, the MTT reagent (Ck = 0.5 mg mL^−1^) was added to a well of a microtiter plate. After 2 h of incubation (37 °C, 5% CO_2_) the MTT solution was removed and replaced with DMSO, to which Sorensen’s buffer (0.1 M glycine, 0.1M NaCl; pH = 10.5) was then added. The cell viability measured in that way was directly proportional to the absorbance (λ = 570 nm). The mean of the absorbance was the amount of viable cells with respect to the control (100%). The control was an un-doped nanohydroxyapatite disc immersed in pure DMEM medium with the addition of FBS. For MTT analysis, three experiments were performed using cells from consecutive passages.

#### 4.3.9. Adhesion Assay

In a 24-well plate, after activation of the tested nanoapatites and nanohydroxyapatites as control (C), discs (DMEM medium with 5% FBS) and Balb/3T3 mouse embryonic fibroblasts in the density of 2 × 10^4^ cells/disk/well were added and incubated for 24 h under CO_2_-enriched conditions. After this time, the ability of the cells to adhere to the tested biomaterials was examined using fluorescence microscopy. The cells were stained in new plates with 1 µL of Calcein AM (Thermo Fisher Scientific, Waltham, QC, Canada) and 1 µL of DAPI (Thermo Fisher Scientific, Waltham, QC, Canada). An analysis of the adhesion area on the surfaces of the tested biomaterials is provided in Section 4.3.11. below.

#### 4.3.10. Influence of Bacterial Biofilm on the Viability of Fibroblast Cells on Nanoapatites

Tested biomaterials with cultures of *S. aureus* or *E. faecalis* (10^5^ CFU mL^−1^) were incubated with fibroblasts at a density of 2 × 10^4^ cells/disc/well. After 24 h incubation under CO_2_-enriched conditions, supernatants from the wells were harvested, and the discs were washed with PBS supplemented with FBS. The percentage of biofilm area and fibroblast viability were measured according to Section 4.3.11. below.

#### 4.3.11. Fluorescence Microscopy (FM) and Computational Analysis of Pictures

The adhered Balb/3T3 mouse embryonic fibroblasts or/and bacterial biofilm on the surface of nanoapatites discs were stained in non-adherent 48-well plates with 1 µL of propidium iodide (Ex λ = 543 nm) and 1 µL SYTO9/ Calcein AM (Ex λ = 488 nm) for 30 min using a LIVE/DEAD BacLight Bacterial Viability Kit (Thermo Fisher Scientific) or Calcein AM (Thermo Fisher Scientific, Canada). Cell nuclei were stained with 1 µL DAPI (Ex λ = 359 nm). Imaging was performed on an Axio Inverted Observer fluorescence microscope 7 (Carl Zeiss, Erbach, Germany) equipped with an Orca Flash 40 camera (Hamamatsu, Hamamatsu city, Japan) and 20× objective Plan APO. Scale bar = 100 µm.

A detailed computer-qualitative and quantitative analysis of the obtained pictures was performed by estimating the coverage area on the examined biomaterials (the percentage of the area occupied by eukaryotic cells or/and bacterial biofilm). The cell vibility analysis was performed by calculating the fluorescence intensity of dyes used. Acquired images were processed and analyzed using Fiji/ImageJ software ver. 1.53c (NIH, Bethesda, MD, USA). First, maximum intensity projections (MIP) were obtained from stacks of images. The areas from binarised images were then transferred onto original live and dead channel MIP images and mean fluorescence intensities of all detected objects per field of view were calculated using ImageJ’s Analyze Particles function [80].

#### 4.3.12. Scanning Electron Microscope (SEM)

Measurement was performed by a modified previous method [81]. Selected bacterial strains of the tested bacteria were added to TSB medium to obtain the final denisity of 10^5^ CFU mL^−1^. Sterile discs of selected nanoapatites doped with noble metals (numbered as: 2, 4, 6, 8, 10, 12, and 14) were added to 48-well non-adherent plates, while non-doped nanohydroxyapatite was used as a control (C). Selected biomaterials were imaged using a scanning electron microscope (SEM, FEI Nova NanoSEM 230, Hillsboro, OR, USA). Magnifications ranging from 20,000 to 50,000× were used. Measuring bars ranged from 1–3 µm as stated in the description of each picture.

#### 4.3.13. Statistical Analysis

Variance analysis was performed using the software Statistica 13 ver. 13.3.721.0 (StatSoft, Tulsa, OK, USA) (ANOVA one-way analysis). One-way ANOVA analysis was performed in the above studies. A probability value of *p* < 0.05 was considered significant.

## 5. Conclusions

In summary, almost all nanoapatites doped with noble metals (Ag, Au, and Pd) exhibit strong anti-adhesive and anti-biofilm properties against *E. faecalis* VRE200 and *S. aureus* MRSA P19, especially in the double- and triple-metallic composition: Ag-Au; Ag-Pd; and Ag-Au-Pd. Most of the mono-metallic nanoapatites were bacteriostatic, while most of the selected double- and triple-metallic components were bactericidal. Metals within triple-metallic component nanoapatites were observed to exhibit synergistic interactions, suggesting that they may have a different mechanism of action against the bacteria tested. Nanoapatites doped with noble metals did not significantly reduce the viability of eukaryotic cells. Furthermore, it was shown that some of them can even positively affect the viability of fibroblasts co-incubated with bacterial biofilms. This research shows the great potential of nanoapatites doped with noble metals to be used in the future in medical sectors.

## Figures and Tables

**Figure 1 ijms-23-01533-f001:**
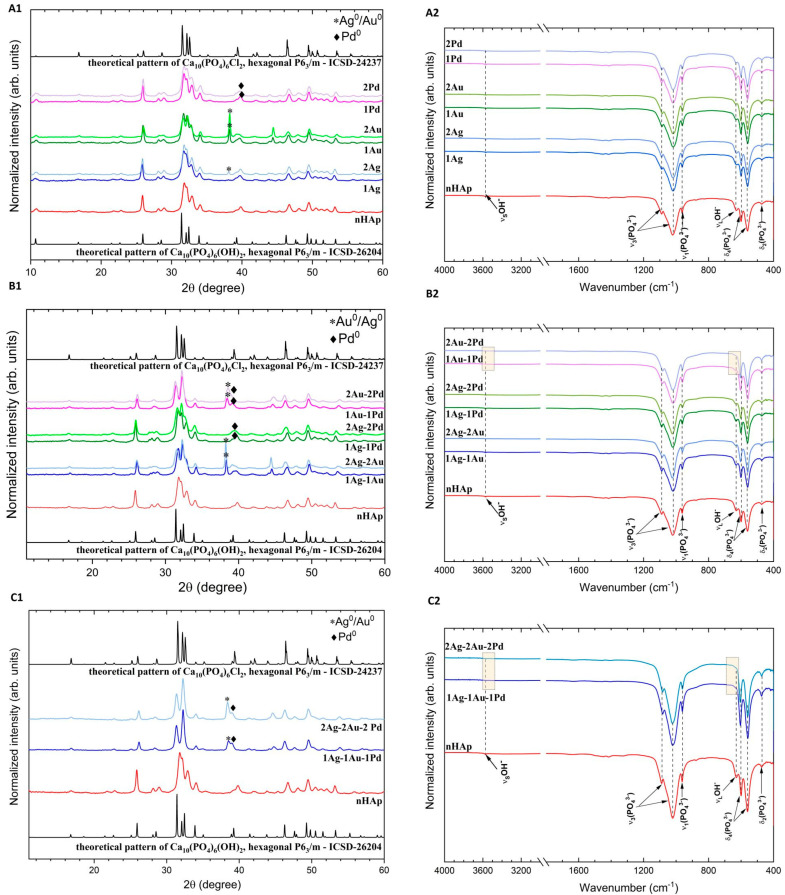
XRPD diffraction patterns of the nanoapatites doped (**A1**), double-doped (**B1**), and triple-doped (**C1**) with Ag^+^, Au^+^, and Pd^2+^ ions at a concentration of 1 mol% and 2 mol%, with the indication of metallic precipitation of noble metals. The diffractograms of these materials were juxtaposed with undoped hydroxyapatite (nHAp). FTIR spectra of the nanoapatites doped (**A2**), co-doped (**B2**), and triple-doped (**C2**) with Ag^+^, Au^+^, and Pd^2+^ ions at a concentration of 1 mol% and 2 mol%. The juxtaposition of the spectra shows the absence of or slightly visible *ν_l_* (OH^−^) band characteristic for hydroxyapatite in the case of materials with a higher share of the chlorapatite phase (nClAp) (marked area). Sample names have been abbreviated, e.g., sample code OH-Cl-Ap: 1 mol% Ag^+^, 1 mol% Au^+^ is 1Ag-1Au.

**Figure 2 ijms-23-01533-f002:**
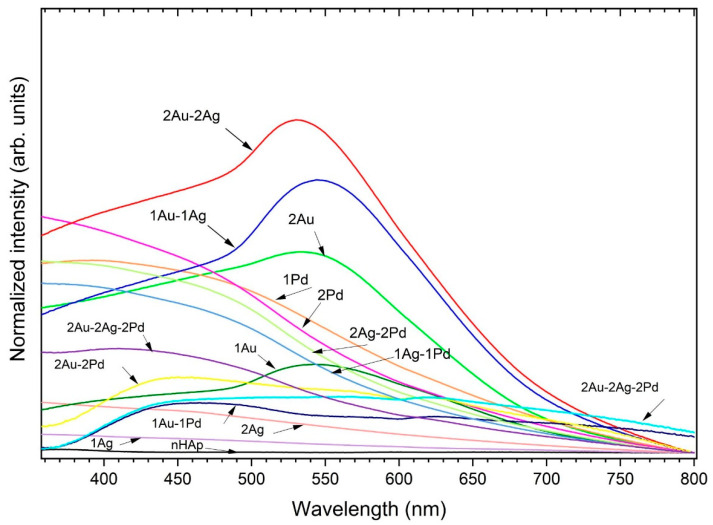
UV–Vis spectra of the nanoapatites doped, double-doped, and triple-doped with Ag^+^, Au^+^, and Pd^2+^ ions at a concentration of 1 mol% and 2 mol% with the indication of plasmon resonance peak (SPR) for metallic noble metals. Sample names have been abbreviated, e.g., sample code OH-Cl-Ap: 1 mol% Ag^+^, 1 mol% Au^+^ is 1Ag-1Au.

**Figure 3 ijms-23-01533-f003:**
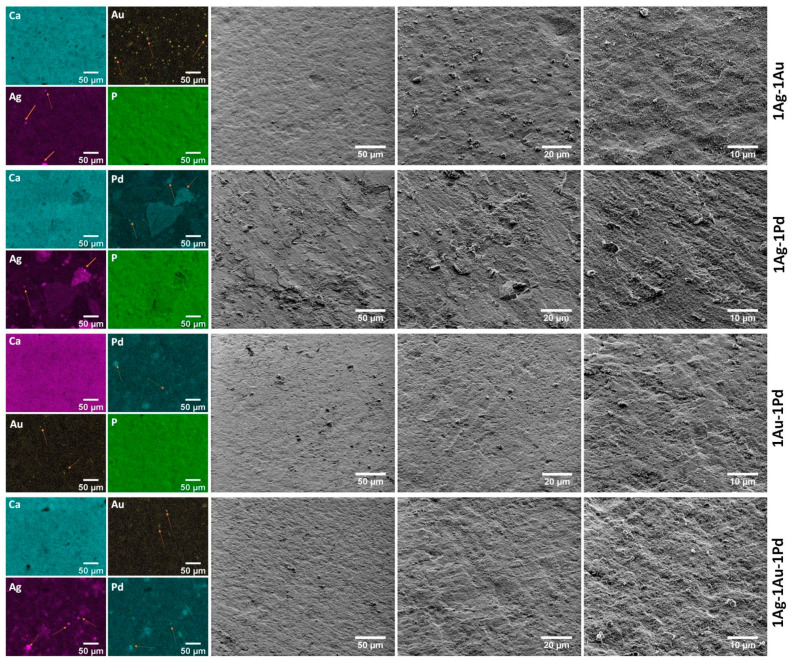
Representative SEM images (50 µm scale bar—1000× magnification, 20 µm—2500×, and 10 µm—5000×) of the nanoapatites pellets double-doped and triple-doped with Ag^+^, Au^+^, and Pd^2+^ ions at a concentration of 1 mol% together with EDS elemental maps. Arrows on the maps indicate metallic precipitates. Sample names have been abbreviated, e.g., sample code OH-Cl-Ap: 1 mol% Ag^+^, 1 mol% Au^+^ is 1Ag-1Au.

**Figure 4 ijms-23-01533-f004:**
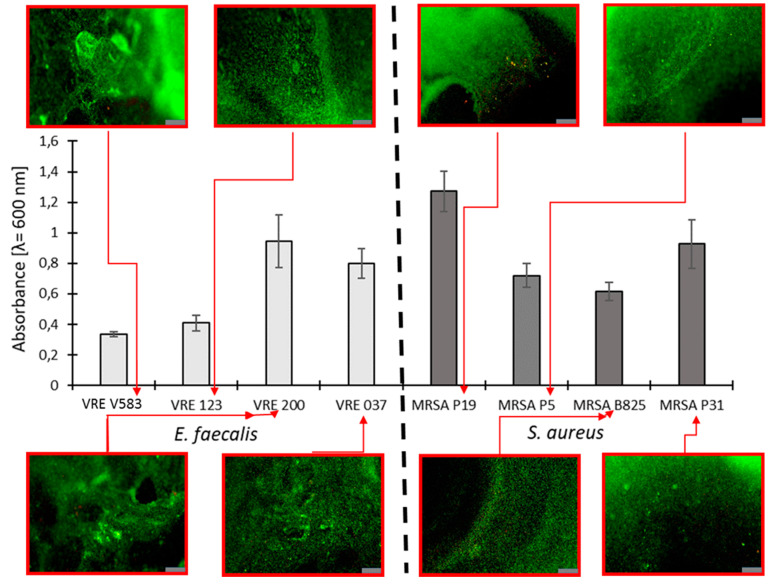
The graph shows the amount of biofilm production by the tested drug-resistant clinical strains *E. faecalis* and *S. aureus* and representative pictures of biofilm formation detected by fluorescence microscopy; Scale bar = 100 µm.

**Figure 5 ijms-23-01533-f005:**
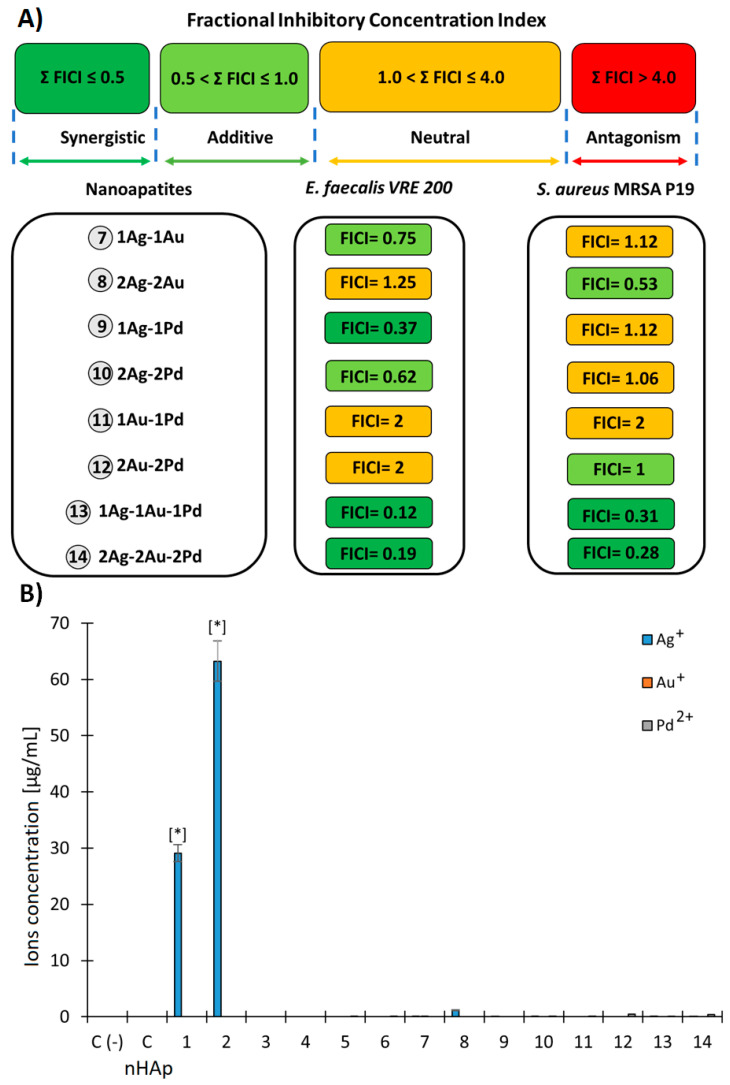
(**A**) Graphical representation of results of antibacterial interactions between the individual dope of noble metals of nanoapatites against *E. faecalis* VRE 200 and *S. aureus* MRSA P19. The interactions were interpreted based on the calculation of the fractional inhibitory concentration index (FICI). (**B**) The release of metal ions (Ag^+^; Au^+^; and Pd^2+^) from the tested nanoapatites. The concentration of metal ions (µg/mL) was measured in the TSB medium at 37 °C after 48 h of incubation. Control negative (C-) was a pure TSB medium; control (C HAp) was a pure nanohydroxyapatite; mean ± SD, n = 3; * statistically different from the control *p* < 0.05.

**Figure 6 ijms-23-01533-f006:**
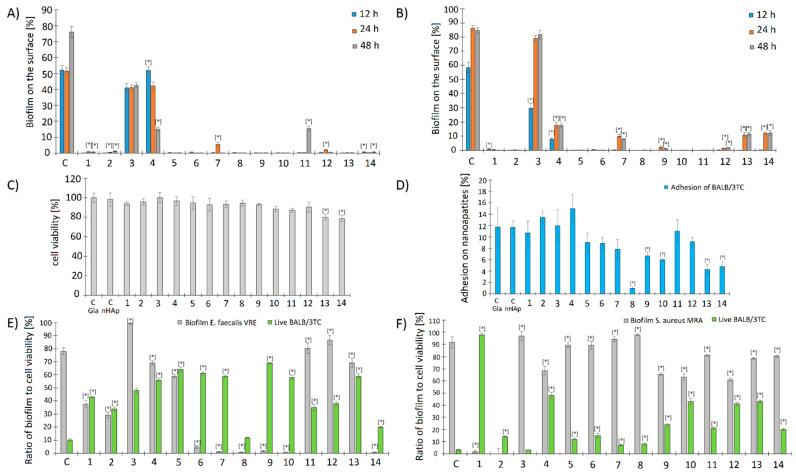
The influence of the studied nanoapatites on the adhesion and biofilm formation by drug-resistant clinical strains of (**A**) *E. faecalis* VRE 200 and (**B**) *S. aureus* MRSA P19 was determined in three incubation periods (12 h, 24 h, and 48 h). The next graphs show the (**C**) cytotoxicity (%) and (**D**) adhesion (%) of Balb/3T3 mouse embryonic fibroblasts to the surface of the tested biomaterials: C HAp-control on nanohydroxyapatite surface and C Gla-control on glass surface. The last graphs show the influence of the biofilm of the tested bacterial strains (**E**) *E. faecalis* VRE 200 and (**F**) *S. aureus* MRSA P19 on the viability of Balb/3T3 mouse embryonic fibroblasts cells [%] adhered on the surface of the tested biomaterials. Control (C) is pure nanohydroxyapatite; mean ± SD, n = 3; * statistically different from the control *p* < 0.05.

**Figure 7 ijms-23-01533-f007:**
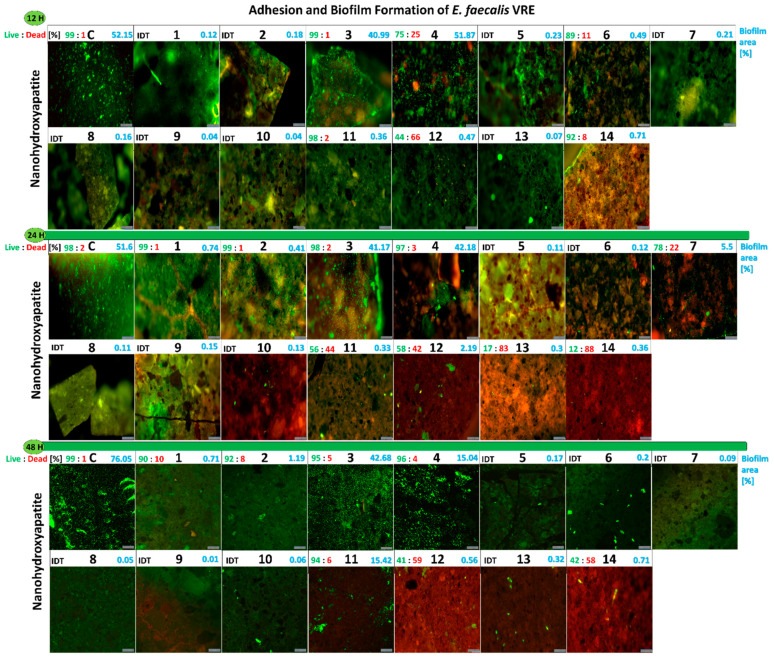
Images from the fluorescence microscopy presenting adhesion and biofilm formation of drug-resistant clinical strains *E. faecalis* VRE 200 on tested nanoapatites doped with noble metals in three incubation points: 12 h (**top**), 24 h (**middle**), and 48 h (**bottom**). Information above the photos: on the left side, the biofilm viability (Live:Dead) (%); on the right side, biofilm area (%) on surfaces of nanoapatites discs. As a control, pure nanohydroxyapatites discs were used. IDT—indeterminate. Scale bars = 100 µm.

**Figure 8 ijms-23-01533-f008:**
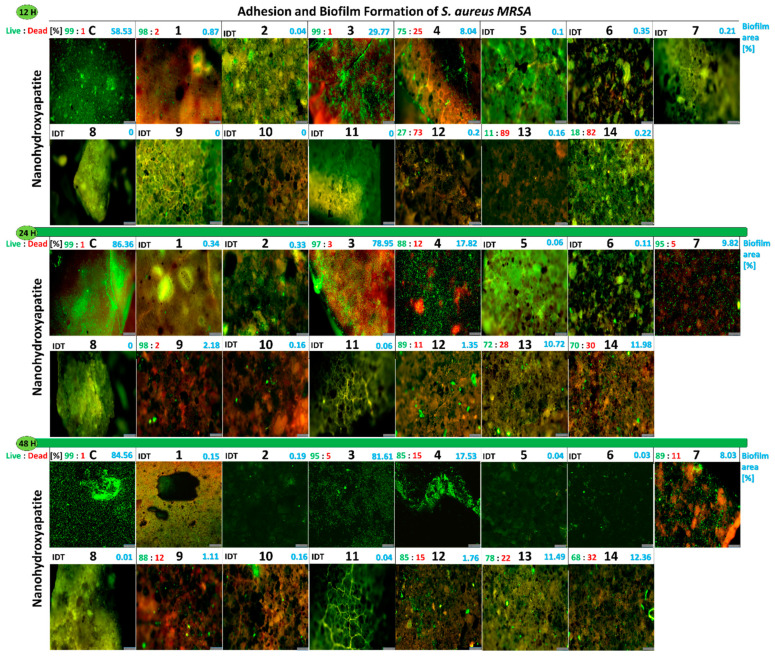
Images from the fluorescence microscopy presenting adhesion and biofilm formation of drug-resistant clinical strains *S. aureus* MRSA P19 on tested nanoapatites doped with noble metals in three incubation points: 12 h (**top**), 24 h (**middle**), and 48 h (**bottom**). Information above the photos: on the left side, the biofilm viability (Live:Dead) [%]; on the right side, biofilm area (%) on surfaces of nanoapatites discs. As a control, pure nanohydroxyapatites discs were used. IDT—indeterminate. Scale bars = 100 µm.

**Figure 9 ijms-23-01533-f009:**
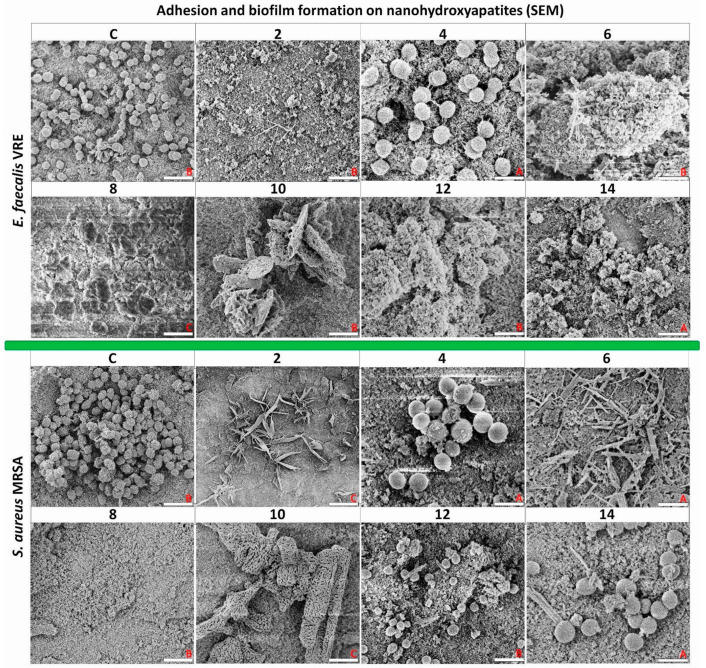
Images obtained by scanning electron microscopy (SEM) visualization demonstrating adhesion and biofilm formation of drug-resistant clinical strains *E. faecalis* VRE 200 (**top**) and *S. aureus* MRSA P19 (**bottom**) on tested nanoapatites doped with noble metals after a 48 h incubation. As a control, pure nanohydroxyapatites discs were used. Details of the magnification (M) and scale bar (sc) are in the description at the bottom of the images: A = M: 50,000×, sc: 1 µm; B = M: 25,000× sc: 2 µm; C = M: 20,000× sc: 3 µm.

**Table 1 ijms-23-01533-t001:** Contents of dopants in the obtained nanoapatites: molar content of Ag^+^, Au^+^, Pd^2+^ ions, ratio of (Ca^2+^+(Ag^+^/Au^+^/Pd^2+^)) to phosphorus (cat./P) ions and the percentage of Cl^−^ in relation to (OH^−^, Cl^−^) anions. Calculations were made on the basis of SEM–EDS measurement data (see tables in the Appendix A, inset). Sample names have been abbreviated, e.g., sample code OH-Cl-Ap: 1 mol% Ag^+^, 1 mol% Au^+^ is 1Ag-1Au.

Abb.	Sample	Ag (mol%)	Au (mol%)	Pd (mol%)	Cl/(Cl + OH) × 100%	cat./P
**C**	nHAp	-	-	-	-	1.64
**1**	1Ag	1.1	-	-	-	1.68
**2**	2Ag	1.8	-	-	-	1.62
**3**	1Au	-	1.0	-	25	1.71
**4**	2Au	-	1.3	-	49	1.71
**5**	1Pd	-	-	1.3	13	1.68
**6**	2Pd	-	-	2.3	23	1.70
**7**	1Ag-1Au	0.8	0.9	-	43	1.69
**8**	2Ag-2Au	1.2	1.9	-	58	1.63
**9**	1Ag-1Pd	0.6	-	0.9	14	1.65
**10**	2Ag-2Pd	1.6	-	1.7	40	1.68
**11**	1Au-1Pd	-	1.0	0.5	73	1.69
**12**	2Au-2Pd	-	2.3	1.8	85	1.70
**13**	1Ag-1Au-1Pd	0.5	0.9	0.8	90	1.69
**14**	2Ag-2Au-2Pd	1.5	1.9	1.8	79	1.72

**Table 2 ijms-23-01533-t002:** Minimum inhibitory concentrations (MIC) and minimum bactericidal concentrations (MBC) of nanoapatites (µg/mL).

Abb.	Sample	*E. faecalis* VRE 200		*S. aureus* MRSA P19	
		MIC	MBC	MIC	MBC
**C**	nHAp	>8192	>8192	>8192	>8192
**1**	1Ag	2048	2048	1024	1024
**2**	2Ag	1024	1024	512	512
**3**	1Au	4096	8192	8192	>8192
**4**	2Au	4096	8192	8192	>8192
**5**	1Pd	4096	8192	8192	>8192
**6**	2Pd	4096	8192	8192	>8192
**7**	1Ag-1Au	1024	1024	1024	1024
**8**	2Ag-2Au	1024	1024	256	256
**9**	1Ag-1Pd	512	512	1024	1024
**10**	2Ag-2Pd	512	512	512	512
**11**	1Au-1Pd	4096	8192	8192	>8192
**12**	2Au-2Pd	4096	8192	4096	>8192
**13**	1Ag-1Au-1Pd	128	128	256	256
**14**	2Ag-2Au-2Pd	128	128	128	128

## Data Availability

The data presented in this study are available on request from the corresponding authors.

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
