# Peer review of "Nanoapatites Doped and Co-Doped with Noble Metal Ions as Modern Antibiofilm Materials for Biomedical Applications against Drug-Resistant Clinical Strains of Enterococcus faecalis VRE and Staphylococcus aureus MRSA"

_ijms, 2022, doi:10.3390/ijms23031533_

Round 1
Reviewer 1 Report
The manuscript by Paluch et al. “Nanoapatites doped and co-doped with noble metal ions as modern antibiofilm materials for biomedical applications against drug-resistant clinical strains of VRE and MRSA” is interesting. There are a few major concerns to be addressed before its publication as follows:
Major comments.
- Introduction, citations are missing for the many sentences containing specific scientific statements. However, the introduction section needs to be more polished with appropriate recent citations. Please include some points about (i) biofilm formation mechanism and its role in drug resistance especially antibiotics (i.e. doi: 10.1016/j.biotechadv.2018.11.007), (ii) detailed mechanism and selectivity and/specificity of nanomaterials as antimicrobial agents towards Gram +ve/-ve microbes, including S. aureus and E. faecalis as Gram+ve. (i.e. doi: 10.1039/C6RA14688K).
- The authors should add an illustration (Fig.) about the mechanism or possible role of synthesized nano-materials as antimicrobials towards S. aureus and E. faecalis. How about its effectiveness as compared to literature (comparison)?
- To many Figures in the main text, Please add up to 8 Figures or Tables in the main text and others can be added as supporting files.
- The discussion section is weak, it can be improved significantly especially, how multi-metals enhance the antimicrobial potentials (a mechanism), physiological changes in the organism, how effective as compared to literature reports? and highlight the significance present study.
- The materials and methods sections can be significantly reduced using appropriate citations as most of the procedures are well-known.
Minor comments
- Add organism name in the title.
- All abbreviations should be cross-checked and to be verified (separately in the abstract and main text).
- The organism name should be in “italics”.
- Please correct “nanamaterial” line 200.
Author Response
We thank for valuable issues which help us improve the manuscript. Changes in the manuscript were marked in yellow. Detailed answers for your suggestion were attached below.
Major comments.
1. Introduction, citations are missing for the many sentences containing specific scientific statements. However, the introduction section needs to be more polished with appropriate recent citations. Please include some points about (i) biofilm formation mechanism and its role in drug resistance especially antibiotics (i.e. doi: 10.1016/j.biotechadv.2018.11.007), (ii) detailed mechanism and selectivity and/specificity of nanomaterials as antimicrobial agents towards Gram +ve/-ve microbes, including S. aureus and E. faecalis as Gram+ve. (i.e. doi: 10.1039/C6RA14688K).
Answer: Major changes were introduced due to your suggestions. In our best knowledge, the newest investigations were included in the introduction section. Moreover, biofilm formation mechanism and its role in drug resistance especially antibiotics as well as detailed mechanism and selectivity of nanomaterials as antimicrobial agents were added to the introduction. We thank you for paying attention to this aspect.
2. The authors should add an illustration (Fig.) about the mechanism or possible role of synthesized nano-materials as antimicrobials towards S. aureus and E. faecalis. How about its effectiveness as compared to literature (comparison)?
Answer: We thank for valuable suggestion. However, manuscript contains a lot of figures. We want to use your idea in the next manuscript (about nanoapatites molecular mechanism of action).
3. To many Figures in the main text, Please add up to 8 Figures or Tables in the main text and others can be added as supporting files.
Answer: Numbers of figures were reduced to 9. We chose the figures which are required to understanding of the manuscript. In our opinion lower amounts of figures may decrease quality of the article.
4. The discussion section is weak, it can be improved significantly especially, how multi-metals enhance the antimicrobial potentials (a mechanism), physiological changes in the organism, how effective as compared to literature reports? and highlight the significance present study.
Answer: Discussion was corrected and supplemented due to your suggestions.
5. The materials and methods sections can be significantly reduced using appropriate citations as most of the procedures are well-known.
Answer: We reduced following section: Assessment of Biofilm Production Ability, Minimal Inhibitory and Bactericidal Concentrations (MIC and MBC), Scanning Electron Microscope (SEM)
Minor comments
1. Add organism name in the title.
Answer: It was corrected.
2. All abbreviations should be cross-checked and to be verified (separately in the abstract and main text).
Answer: It was corrected.
3. The organism name should be in “italics”.
Answer: It was corrected.
4. Please correct “nanamaterial” line 200.
Answer: It was corrected
Reviewer 2 Report
The manuscript presented for review is a report about the nanoapatites doped and co-doped with noble metal ions as modern antibiofilm materials for biomedical applications against drug-resistant clinical strains of VRE and MRSA. Multidrug resistance strains pose a serious treatment problem. Therefore, the subject of presented manuscript raises an especially important clinical problem. The research was conducted in a very detailed way and the results are clearly presented. The manuscript is interesting, complete, and well structured, and it is necessary to do only a minor revision to be accepted:
Minor specific suggestion/comments:
Line 33: haven’t à have not
Line 45: antibiotics and disinfectants à antibiotics, and disinfectants
Line 46: e.g.,teeth à e.g., teeth
Line 46: orthopaedic à orthopedic
Line 61: All of above à All above
Line 64: bioactivity and à bioactivity, and
Line 70: mutlidrug à multidrug
Line 465: patients à patient’s
Line 468: properties,in à properties, in
Line 623: parragraph
Genre names should be italicized throughout the text e.g., line 598
Author Response
We thank you for the suggestions. Minor changes was provided to the manuscript. Changes were marked in yellow.
Reviewer 3 Report
The paper presents an analysis of antibacterial properties of three different metals in nHAp. The paper is interesting and thorough and certainly publishable.
"This suggests that the silver released into the medium 268
may increase the antimicrobial effect, but only for one-component nanoapatites(figure 8)" is factually untrue. What is suggested is that silver produces ions and alloying prevents the production of ions. Figure 8 is providing clear evidence that the ions are forming some alloy nanoparticles which is consistent with the synergy seen in figure 7. Antibacterial discussion should begin with bacteria in the dataset.
Author Response
We thank you for paying attention to this aspect. Changes in the manuscript were marked in yellow. Manuscript was corrected due to your suggestions. Introduced changes was presented below:
“The results of the release of metal ions (Ag+, Au+, Pd2+) from the tested biomaterials at various incubation times show a significant increase in the amount of Ag+ ions with silver doped and co-doped nanoapatites (Figure 5B). In particular for biomaterial 1Ag (1) and and 2Ag (2) released nearly 30 µg/mL and 63 µg/mL of Ag+ ions, respectively. Other metallic elements were only detected in trace amounts. This suggests that the silver released into the medium may increase the antimicrobial effect (Figure 5B).”
Moreover, discussion was corrected and expanded.
Reviewer 4 Report
Even though the study of nanocrystalline apatites doped and co-doped with noble metal ions as potential anti-adhesive and anti-biofilm agents are new, the most of them contained an additional phase of metallic nanoparticles were confirmed by the XRPD, FT-IR, UV-Vis and SEM-EDS techniques, and extensive microbiological tests of the nanoapatites were carried out, however, they showed very common MIC, MBC values, despite the various testing and characterization made by the author. Related publications about antibacterial activity and the detail mechanism illuminate haven’t been cited and discussed, such as Polyhedron, 2018, 139, 296; Inorg. Chim. Acta, 2018, 473, 112, etc; In my opinion, this manuscript is not qualified.
Author Response
We thank you for the suggestions. Major changes was provided to the introduction part of the manuscript. In our best knowledge, the newest investigations were included in the introduction section. Discussion was significantly expanded due to your issues. Suggested articles were included in the present version of the manuscript. Changes in the manuscript were marked in yellow.
Round 2
Reviewer 1 Report
Accept as is.